# Exploiting the Potential of Magnetic Nanoparticles for Rapid Diagnosis Tests (RDTs): Nanoparticle-Antibody Conjugates and Color Development Strategies

**DOI:** 10.3390/diagnostics13193033

**Published:** 2023-09-23

**Authors:** Yeonjeong Ha

**Affiliations:** ICT Environment Convergence, Department of ICT Convergence, College of IT Engineering, Pyeongtaek University, 3825 Seodong-daero, Pyeongtaek-si 17869, Gyeonggi-do, Republic of Korea; yha@ptu.ac.kr

**Keywords:** magnetic nanoparticles (MNPs), nanoparticle-antibody conjugates, immunoassay, rapid diagnosis tests (RDTs)

## Abstract

Magnetic nanoparticles (MNPs) have emerged as a promising material in disease diagnostics due to their potential to enhance detection sensitivity, facilitate concentration and purification of target substances in diverse samples, and enable favorable color-based detection. In this study, antibody-conjugated MNPs were successfully synthesized and validated through two appropriate methods: the measurement of MNPs’ size and the use of phosphatase methods. Additionally, three methods were suggested and implemented for developing color in MNPs-based immunoassay, including the formation of MNP aggregations, utilization of MNPs’ peroxidase-like activity, and synthesis of dually-conjugated MNPs with both enzyme and antibody. In particular, color development utilizing nanoparticle aggregations was demonstrated to result in a more yellowish color as virus concentration increased, while the peroxidase activity of MNPs exhibited a proportional increase in color intensity as the MNP concentration increased. This observation suggests the potential applicability of quantitative analysis using these methods. Furthermore, effective concentration and purification of target substances were demonstrated through the collection of MNPs using an external magnetic field, irrespective of factors such as antibody conjugation, dispersion medium, or virus binding. Finally, based on the key findings of this study, a design proposal for MNPs-based immunoassay is presented. Overall, MNPs-based immunoassays hold significant potential for advancing disease diagnostics.

## 1. Introduction

In the present era, we are encountering various new diseases. Since the COVID-19 pandemic, there has been an imperative to accurately and quickly diagnose diseases caused by various factors such as viruses, bacteria, and food allergens [1,2,3]. Although reverse transcription polymerase chain reaction (RT-PCR) currently enables accurate disease diagnosis [4,5], its drawback lies in the considerable time investment required [6]. On the other hand, lateral flow immunoassay, which utilizes membranes in rapid detection kits, can provide quick results but has the disadvantage of lower accuracy [7,8,9]. Therefore, there is a crucial need for the development of diagnostic methods that are both rapid and accurate. This is particularly important in developing countries where accessibility to healthcare professionals is limited. Additionally, it is essential to develop diagnostic kits that can withstand high temperatures, high humidity, and long transportation times while maintaining stability.

As part of the concerted efforts to improve rapid detection kits, magnetic nanoparticles (MNPs) are regarded as attractive tools for a broad range of applications, including accurate and rapid diagnosis due to their unique properties [7,10,11]. Firstly, MNPs can be easily and rapidly collected using external magnetic forces [12,13], enabling effective concentration and purification of target substances. This capability effectively enhances the sensitivity of diagnostic kits. Furthermore, MNPs can efficiently conjugate with various biomaterials, including antibodies [14,15,16,17], and their small size allows for rapid contact with target substances, enabling swift diagnostics [18,19]. Indeed, MNPs have been successfully utilized for the rapid and accurate detection of various biological pathogens (e.g., viruses and bacteria) [20,21,22,23,24,25,26] as well as environmental pollutants [27,28,29] in previous studies. However, as of now, there is a lack of consensus on the standard methods for synthesizing antibody-conjugated MNPs and confirmation methods for conjugating antibodies in various research studies. Additionally, in the case of diagnostic kits designed for use by the general public, the requirement for color expression in the diagnostic process is essential. Therefore, a thorough investigation is necessary to explore diverse approaches that enable color expression in MNPs-based diagnostics to be used as an alternative to complex devices that may pose challenges for non-professionals.

In this study, antibody-conjugated MNPs were synthesized, and the conjugation of antibodies to MNPs was confirmed by measuring the size of MNPs and utilizing the phosphatase method. Additionally, three methods for color development in a rapid detection kit using MNPs were performed, and strategies for their application in a rapid diagnostic test (RDT) were discussed. Furthermore, it was demonstrated that MNPs can be easily collected using an external magnetic field, regardless of antibody conjugation, dispersion medium, or virus binding, to concentrate and purify target substances. Lastly, the key findings from this study were used to design an RDT, providing clues for the future development of MNPs-based immunoassays.

## 2. Materials and Methods

### 2.1. Materials

Carboxymethyl-dextran-coated 100 nm MNPs were purchased from Chemicell (Berlin, Germany), and carboxylic acid functionalized 50 nm MNPs were purchased from Sigma Aldrich (St. Louis, MO, USA). Human Coxsackievirus B3 (COX3) virus and Enterovirus (EV71) virus were kindly provided by the Korea Bank for Pathogenic Viruses. Two types of antibodies, Coxsackievirus B3 antibody (COX3Ab) and Enterovirus 71 antibody (EV71Ab), were purchased from Sigma Aldrich (St. Louis, MO, USA). A magnetic separator (DYNAMAG-2), alkaline phosphatase antibody, and horse radish peroxidase (HRP) were purchased from Thermo Fisher Scientific (Waltham, MA, USA). All other chemicals were obtained from Sigma Aldrich (St. Louis, MO, USA) with high purity (>99.9%).

### 2.2. Synthesis and Validation of Antibody-Conjugated MNPs

The immobilization of antibodies on MNPs coated with carboxymethyl-dextran was carried out using N-(3-dimethylaminopropyl)-N’-ethylcarbodiimide hydrochloride/N-hydroxysuccinimide (EDC/NHS) chemistry, as described in previous studies [15,30]. Initially, 50 μL of a 25 mg mL^−1^ solution of MNPs was transferred into a 1.5 mL centrifuge tube. Subsequently, 2.5 μL of 0.4 M EDC, 50 μL of 0.1 M NHS, and 450 μL of ×1 phosphate buffer saline (PBS) were added. The centrifuge tube was then incubated at 37 °C and 200 rpm for a duration of 30 min. The MNPs activated using EDC/NHS chemistry were separated using a magnetic separator, and the supernatant was removed. The MNPs were then resuspended in 450 μL of PBS for washing. This washing step was repeated three times. Next, 10 μg of each antibody (COX3Ab or EV71Ab) was added to the tube and incubated at 37 °C and 200 rpm for 2 h, followed by three rounds of washing. To block any remaining inactive surface, 1% bovine serum albumin (BSA) was added. After removing the supernatant using a magnetic separator, the MNPs were carefully washed with PBS more than 10 times. The final suspension of antibody-conjugated MNPs in PBS was stored at 4 °C until further use. In cases of prolonged storage of MNPs–antibody conjugations, the formation of large aggregations may impact stability. Therefore, the conjugations were utilized for experiments within one week after synthesis. 

To validate the antibody-conjugated MNPs, first, a comparison of the size of MNPs before and after antibody conjugation was performed using dynamic light scattering (DLS) analysis conducted with a Malvern Zetasizer (Westborough, MA, USA). For every measurement, I prepared one sample each (MNPs only, antibody-conjugated MNPs, such as MNPs-EV71Ab and MNPs-COX3Ab) and conducted three measurements for each sample to verify instrumental measurement errors. Secondly, the phosphatase method involving the use of an alkaline phosphatase secondary antibody was conducted. Both bare MNPs and antibody-conjugated MNPs (MNPs-COX3Ab and MNPs-EV71Ab) were incubated with 10 μg of the phosphatase secondary antibody at 37 °C and 200 rpm for a duration of 2 h. Subsequently, the MNPs were washed multiple times until the supernatant exhibited no color development upon the addition of p-nitrophenyl phosphate solution. Following this, bare nanoparticles (MNPs only), bare MNPs, and antibody-conjugated MNPs incubated with the phosphatase secondary antibody (referred to as MNPs-P and MNPs-COX3Ab-P (or MNPs-EV71-P), respectively) were transferred into 96-well plates at varying volumes (5 μL, 10 μL, and 20 μL). Next, p-nitrophenyl phosphate solution was added to each well, and a waiting period of 30 min was observed for color development. Subsequently, the absorbance at 405 nm was measured using a Hidex microplate reader (Hidex, Turku, Finland).

### 2.3. Determination of Peroxidase-like Activity of MNPs

To investigate the peroxidase-like activity of MNPs, various concentrations of MNPs with a size of 100 nm ranging from 0 to 3.5 mg mL^−1^ were prepared. A stock solution of tetramethylbenzidine (TMB) at a concentration of 1 mg mL^−1^ was prepared by dissolving the chemical in dimethyl sulfoxide (DMSO). Subsequently, 1 mL of the TMB stock solution was diluted with 9 mL of a 0.2 M sodium acetate buffer solution, resulting in the formation of the TMB substrate solution. To determine the optimal concentration of hydrogen peroxide (H_2_O_2_) for color development induced by the peroxidase-like activity of MNPs, different volumes (2 μL, 10 μL, 1 mL, and 2 mL) of a 30% H_2_O_2_ solution were added to the TMB substrate solution. The TMB substrate solution containing a specific volume of 30% H_2_O_2_ solution was then added to the dispersions of MNPs in a centrifuge tube. After a 30-min incubation period, the centrifuge tube was gently mixed, and the resulting dispersion was transferred to a 96-well plate. Subsequently, the absorbance at 650 nm was measured using a Hidex microplate reader (Hidex, Turku, Finland).

Following the determination of the optimal volume of 30% H_2_O_2_ (10 μL) for color development, dispersions of bare MNPs with sizes of 50 nm and 100 nm, as well as antibody-conjugated MNPs (MNPs-EV71Ab and MNPs-COX3Ab) with a size of 100 nm, were prepared at different concentrations. These dispersions were then added to the TMB substrate solution containing the previously optimized volume of H_2_O_2_, and the solutions were incubated for 30 min. Subsequently, the absorbance at 650 nm of each solution was measured using a Hidex microplate reader (Hidex, Turku, Finland) to evaluate the peroxidase-like activity.

### 2.4. Synthesis of Dually-Coated MNPs with HRP and Antibody

After activating the MNPs using EDC/NHS chemistry, 10 μg of antibody and horseradish peroxidase (HRP) were added to the MNPs dispersions. The mixture was gently incubated at 37 °C and 200 rpm for a duration of 2 h. Subsequently, the dispersions were subjected to several washes until the supernatant showed no color development upon the addition of the TMB substrate solution containing H_2_O_2_. The binding of HRP onto the MNPs was confirmed by adding the TMB substrate solution with H_2_O_2_ and measuring the absorbance at 650 nm. The antibody binding was also verified using the phosphatase method, as described in Section 2.3.

### 2.5. Virus Binding to Antibody-Conjugated MNPs

To obtain the desired concentration of Human Coxsackievirus B3 (COX3) virus and Enterovirus (EV71) virus, a serial dilution was performed starting from an initial concentration of 4.5 × 10^6^ plaque-forming units (pfu) mL^−1^. In a 96-well plate, each well was coated with 10 μg of capture antibody, followed by an overnight incubation. After washing the plate three times with phosphate-buffered saline (PBS), 100 μL of virus suspensions were added and allowed to incubate at 37 °C for 30 min, facilitating the binding of the virus to the capture antibody. Subsequent washes were performed to remove unbound material.

To assess virus binding to the antibody-conjugated MNPs and the occurrence of MNP aggregations, dispersions of bare MNPs and antibody-conjugated MNPs (MNPs-COX3Ab or MNPs-EV71Ab) were added to the plate and incubated at 37 °C for 30 min. This step allowed for the examination of virus binding to the antibody-conjugated MNPs and the observation of any potential aggregations of MNPs.

## 3. Results 

### 3.1. Confirmation of Antibody-Conjugated MNPs

It is essential to confirm the binding of antibodies to the MNPs for the application of antibody-conjugated nanoparticles in diagnostic kits. In order to confirm the conjugation, two methods were used: (i) assessment of nanoparticle size alteration following the conjugation process and (ii) utilization of the phosphatase method.

#### 3.1.1. Analysis of the Size of Antibody-MNP Conjugates

As shown in Figure 1, dynamic light scattering (DLS) analysis revealed that the size of MNPs conjugated with two types of antibodies (MNPs-EV71Ab:217.73 ± 1.53 nm, MNPs-COX3Ab: 220.70 ± 4.61 nm) was approximately 20 nm larger compared to the size of unconjugated MNPs (190.40 ± 1.05 nm). According to the Student’s *t*-test, the change in size due to antibody conjugation is statistically significant (*p* < 0.05). It should be noted that the actual size of the dextran-coated MNPs used was 100 nm; however, during antibody conjugation, the presence of phosphate buffer saline (PBS) resulted in aggregation, causing an apparent increase in the size of the nanoparticles by approximately 90 nm. MNP aggregation exhibited a yellow coloration, which is advantageous for color expression in the actual diagnosis kit. Therefore, further consideration of the dispersion condition of the MNPs was no longer deemed necessary. Due to the formation of hydration shells around the MNPs, which typically range from a few to tens of nanometers, DLS measurement data may overestimate the size of MNPs.

#### 3.1.2. Confirmation of the Synthesis of Antibody-MNP Conjugates Using Alkaline Phosphatase Antibody 

To further confirm the conjugation of antibodies, the phosphatase method was used [31]. The mechanism of the phosphatase method involves the yellowish color expression in the presence of p-nitrophenyl solution when MNPs and antibodies bind, followed by binding of the antibodies on MNPs to alkaline phosphatase secondary antibodies. Therefore, in the phosphatase method, when MNPs were conjugated with antibodies, an increase in UV/Vis absorbance at 405 nm was observed. As shown in Figure 2, it is evident that MNPs conjugated with EV71Ab and COX3Ab exhibited distinct color expression when bound to an alkaline phosphatase antibody, compared to the binding of the unconjugated MNPs with alkaline phosphatase antibody. Furthermore, by measuring the UV/Vis absorbance at 405 nm, it was confirmed that the color expression increased with the increase in the number of antibody-synthesized nanoparticles. Thus, the successful synthesis of MNPs-antibody conjugation was validated.

### 3.2. Strategies for Developing Color in RDT Kits Using MNPs

In order to apply a rapid diagnosis kit to the general public, it is necessary to develop a color expression that can be visually observed. In this session, research results on three methods for achieving color using MNPs were introduced: (i) color expression resulted from the aggregation of MNPs, (ii) color expression utilizing the peroxidase properties of MNPs themselves, and (iii) simultaneous synthesis of HRP and antibodies on MNPs. Additionally, the methods for applying these findings to the diagnosis kit were discussed.

#### 3.2.1. Color Development Resulting from Aggregations of MNPs

Unlike other commonly used RDTs that use captured antibodies to detect target antigens, when applying antibody-conjugated MNPs to the RDTs, the antigen-antibody binding results in the formation of yellowish aggregation, which can serve as a great strategy for rapid and straightforward visual detection. It is generally acknowledged that the coloration of nanoparticles stems from either the absorption of light by the particles or the scattering of light through nanostructures [32,33]. As shown in Figure 3a, upon capturing the target virus by the capture antibody, the antibody-conjugated MNPs bind to the target virus, facilitating the sedimentation of the MNPs. This results in the manifestation of a yellow color at the bottom. In Figure 3b, #1 depicts the fact that unconjugated MNPs remained dispersed along with the virus (COX3 virus concentration: 4.5 × 10^6^ pfu mL^−1^). However, when using antibody-conjugated nanoparticles, the sedimentation of nanoparticles to the bottom was observed upon their binding with viruses. Furthermore, as the virus concentration increases, more nanoparticles settle at the bottom, creating a denser yellow color (Figure 3b #2, #3). 

This strategy for color development has the potential to augment the precision and sensitivity of the established conventional lateral flow immunoassay (LFIA). This enhancement arises from its capacity to pre-concentrate and pre-purify target analyte samples that are bound to MNPs. In addition, upon the conjugation of MNPs with target material and subsequent binding to capture antibodies, rapid aggregation of MNPs occurs, leading to a pronounced yellowish coloration. This characteristic significantly simplifies the LFIA detection process. Indeed, previous studies successfully reduced the visual detection limits of various target materials, such as fish allergens [34] and viruses [35]. The specific approach to realizing LFIA utilizing MNPs is elaborated in detail in the subsequent Section 4. 

#### 3.2.2. Peroxidase-like Activity of MNPs

In enzyme-linked immunosorbent assay (ELISA), commonly used for the detection of various target substances, including viruses, bacteria, and food allergens, enzymes such as horseradish peroxidase (HRP) have been used for color development based on catalytic activity. However, enzymes have disadvantages for application in rapid diagnostic tests (RDT) due to their susceptibility to changes over time and their vulnerability to destruction at high temperatures. In 2007, Yan’s group made an initial discovery that revealed the intrinsic peroxidase-like activity of MNPs [36], as shown in Figure 4a. This activity arises from the presence of Fe^2+^/Fe^3+^ ions on the MNPs’ surfaces, enabling them to catalyze the decomposition of hydrogen peroxidase. Due to nanoparticles’ enhanced stability under harsh conditions such as high temperature and pressure, the peroxidase-like activity exhibited by MNPs can effectively replace the enzymes traditionally utilized in RDTs. Indeed, previous studies have successfully detected a wide range of target substances by harnessing the peroxidase-like activity of MNPs [37,38,39].

In this study, the conditions for color development using the peroxidase-like activity of 100 nm dextran-coated MNPs were optimized. Different volumes of 30% H_2_O_2_ solution were added to the TMB substrate solution, which was prepared by diluting 1 mL of 1 mg mL^−1^ TMB stock solution in DMSO with 9 mL of 0.2 M sodium acetate buffer solution (Appendix A). The results show that an increase in the concentration of MNPs led to an increase in color development across all ranges of H_2_O_2_ solution (2 μL to 2 mL of 30% H_2_O_2_ solution), clearly demonstrating the intrinsic peroxidase-like activity of MNPs. As depicted in Appendix A, when 10 μL of 30% H_2_O_2_ solution was used, the color development was optimized. Furthermore, under the optimized conditions, the 650 nm absorbance was directly proportional to the concentration of MNPs in the concentration range of 0 to 3.5 mg mL^−1^ (Appendix A). This is because as the number of nanoparticles increases, a greater amount of the oxidation form of TMB is generated. 

Using the optimized conditions, as observed in Figure 4b, the peroxidase-like activity was confirmed not only in 100 nm MNPs but also in 50 nm MNPs. Moreover, it was observed that the color development increased with an increase in the concentration of MNPs. In addition, when 100 nm MNPs were conjugated with two different types of virus antibodies (COX3Ab and EV71Ab), the peroxidase-like activity of MNPs was sustained, and it was observed that the color development increased linearly with an increase in the concentration of nanoparticles. These experimental results demonstrate that antibody-conjugated MNPs can serve as a substitute for enzyme-based color development, highlighting their potential for efficient applications. 

Although the conventional ELISA method has found widespread use for detecting diverse biomaterials, there is a necessity to address its main limitations, which encompass complex procedural steps, prolonged incubation durations, and a relatively high detection limit. The modified ELISA, utilizing the peroxidase-like activity of MNPs, can effectively alleviate the drawbacks of conventional ELISA due to rapid color expression without the need for enzyme and fast antibody-antigen binding facilitated by the high surface area and Brownian motion of MNPs. Furthermore, the incorporation of stable MNPs enhances the reliability of quantitatively detecting target materials in comparison to the conventional enzyme-based ELISA. Indeed, Yang et al. (2013) [37] achieved a substantial reduction in the detection limit for rotavirus when compared to the conventional ELISA method. Similarly, Woo et al. (2013) [40] reduced the detection time of human breast cells by leveraging the peroxidase-like activity of MNPs.

#### 3.2.3. MNP Conjugating with Horseradish Peroxidase (HRP) and Antibodies Simultaneously

The utilization of MNPs in the classical ELISA necessitates the use of a minimum of two antibodies, typically an antibody conjugated with MNPs and a secondary antibody with enzymes of fluorescent dyes. This conventional approach involves multiple sequential steps, resulting in a relatively prolonged detection time (typically exceeding several hours) and rendering the method complex for point-of-care detection. To address this challenge, a novel approach incorporating dually-labeled MNPs was introduced. This innovative strategy entails the concurrent conjugation of both the antibody and enzyme (e.g., HRP) onto the MNPs, as shown in Figure 5a. 

Here, 100 nm MNPs were dually coated with COX3 antibodies and HRP simultaneously. Then, the TMB substrate solution, including H_2_O_2,_ and the phosphatase method were used, respectively, to confirm the successful conjugation of HRP and antibody to the MNPs. Evidently, the UV absorbance at 650 nm of the dually-coated MNPs was considerably higher than that of the negative control (bare MNPs) upon the addition of TMB substrate solution with H_2_O_2_ (Figure 5b). Additionally, the UV absorbance at 405 nm, as determined by the phosphatase method, demonstrated the presence of antibodies on the surface of the dually-coated MNPs (Figure 5c). These results show that both antibody and enzyme stably conjugated with MNPs, exhibiting rapid detection of target materials, can be achieved by capturing and reporting the substances simultaneously by applying dually-coated MNPs to RDTs. The realization of color development through the peroxidase-like activity of MNPs can be considered an effective strategy due to the utilization of inherently stable MNPs without the requirement for enzymes, along with the additional capability of enabling quantitative measurements.

### 3.3. Concentration and Purification of Target Materials Using MNPs

Since the MNPs can act as “nano magnets”, they can easily be collected by applying an external magnet. This is a significant advantage, particularly for rapid diagnostic tests (RDTs) and other biomedical assays, as it simplifies the purification and concentration process. Generally, purification and enrichment of diluted MNP samples in a biological medium (e.g., blood, saliva, etc.) can be performed using three simple steps: (i) collect the MNPs by applying an external magnet field, (ii) remove the supernatant and wash several times, and (iii) resuspend the MNPs in a small volume of sample media. Consequently, the signal-to-noise ratio and detection sensitivity of the assay are improved, leading to more accurate and reliable results.

In this study, it was observed that 100 nm MNPs can be easily collected using an external magnet field at various stages: prior to antibody conjugation (MNPs only), after antibody conjugation (MNP-EV71Ab), and after virus binding to the antibody-conjugated MNPs (MNPs-COXAb-COX3virus) (Figure 6a). Additionally, when dispersed in an artificial saliva solution, the MNPs showed similar behavior to those dispersed in PBS, collecting within 30 min when exposed to an external magnet (Figure 6b). Hence, it was confirmed that when target substances such as viruses and bacteria bind to the antibody-conjugated MNPs within a viscous biological medium (e.g., saliva), they can be easily concentrated and purified solely by applying an external magnetic field without the need for additional steps. 

## 4. Discussion

Through the utilization of key findings in this study, including the conjugation of antibodies to MNPs, the color development methods using MNPs, and the implementation of a concentration/purification technique, it becomes feasible to conceive a diagnostic kit capable of simultaneously concentrating and detecting target substances. Figure 7 illustrates the schematic diagram of a lateral flow immunoassay utilizing MNPs. The diagnostic method can be divided into two main steps. The first step involves the concentration and purification of the target material, where the patient’s sample is incubated with antibody-conjugated nanoparticles, and the sample is subsequently extracted using an external magnetic field. The second step entails applying the concentrated solution into a prepared lateral flow immunoassay (LFI) membrane to detect the target substances (Figure 7). When implementing an MNPs-based lateral flow immunoassay (LFI), the suitable color development method discussed earlier involves color development resulting from aggregations of MNPs. Additionally, it is possible to use MNPs conjugated with fluorescent dye-labeled antibodies for visible detection. Moreover, in cases where a disease can be caused by multiple substances, it is possible to identify the specific substance responsible for the patient’s disease by using nanoparticles coated with antibodies targeting each individual target substance. For example, hand, foot, and mouth disease (HFMD) is primarily caused by two main viruses, Coxsackievirus A16 (COX16) and Enterovirus 71 (EV71). It is known that the EV71 virus is associated with a higher occurrence of neurological complications. By utilizing nanoparticles coated with antibodies specific to COX16 and EV71, it is possible to differentiate the causative virus of the disease, allowing for early treatment of complications.

Conventional enzyme-linked immunosorbent assay (ELISA) can also be modified by incorporating MNPs, as described earlier. Significant enhancements in detection limits and reduction in analysis time can be achieved through the utilization of an MNPs-based ELISA, attributed to the high surface-to-volume ratio and the magnetophoresis (i.e., easy collection by an external magnet) of MNP dispersion. The color development of an MNPs-based ELISA can be achieved through the intrinsic peroxidase-like activity of MNPs and the utilization of dually-labeled MNPs coated with both an enzyme and an antibody. One limitation when applying 100 nm MNPs to ELISA is the potential for the detachment of MNPs from the plate surface during washing steps due to MNPs’ relatively large size. Therefore, it is essential to consider and implement measures to address this issue.

Such MNPs-based LFI have been successfully developed in previous studies as well. For instance, in the study by Yin et al. (2022) [41], nanoparticle aggregation was utilized to detect peanut allergen Ara h1 in MNPs-based LFI, achieving low detection limits not only in phosphate buffer saline (PBS) but also in other media such as milk and chocolate. Furthermore, various substances, including viruses and mycotoxins, have also been detected using MNPs-based LFI [35,42]. 

## 5. Conclusions

Research on rapid detection tests for diseases caused by various biological and chemical factors is rapidly advancing due to its significant importance. However, there is still a need for efforts to achieve faster and more accurate detection. Particularly, in order to detect pathogens in urine or saliva through non-invasive methods that do not involve blood collection, the development of highly sensitive detection methods is required. Immunoassays based on MNPs are considered attractive methods to increase the sensitivity of detection due to the unique properties of MNPs, such as high surface-to-volume ratio, magnetic behavior, and easy biocompatibility. 

In this study, two different antibodies (COX3Ab and EV71Ab) were successfully conjugated to 100 nm MNPs, and confirmation methods for antibody conjugation were proposed. This contribution can be helpful in future research for establishing protocols for the synthesis and confirmation of antibody-conjugated MNPs. In addition, the mechanisms and feasibilities of different methods for color development of MNPs applied to immunoassays were demonstrated. Each color development method can be properly applied to lateral flow immunoassays or modified MNPs-based ELISA methods. Moreover, it was shown that the collection of MNPs using an external magnetic field is straightforward, irrespective of antibody conjugation, dispersion medium, or virus binding. Finally, based on the results of this study, a design for an MNPs-based immunoassay was suggested. 

Considering the increased sensitivity in detection, the potential for concentration and purification of target substances in various samples, and the advantages of color development in detection, MNPs are deemed highly attractive in disease diagnostics at their current stage of development. MNPs are believed to address a variety of challenges in disease diagnostics that need improvement, such as diagnostics in developing countries, non-invasive testing, and accurate testing accessible to the general population.

## Figures and Tables

**Figure 1 diagnostics-13-03033-f001:**
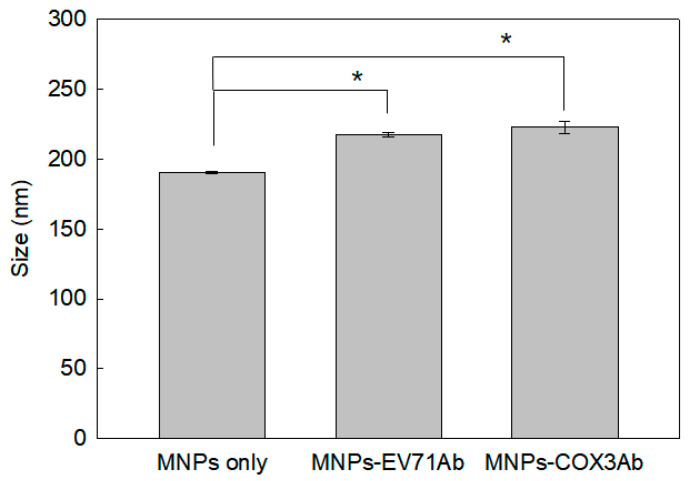
Size of MNPs and MNPs conjugated with antibodies, EV71Ab and COX3Ab, in PBS buffer; * denotes a significant increase in the size of antibody-conjugated MNPs compared to bare nanoparticles, as determined by a Student’s *t*-test (*p* < 0.05).

**Figure 2 diagnostics-13-03033-f002:**
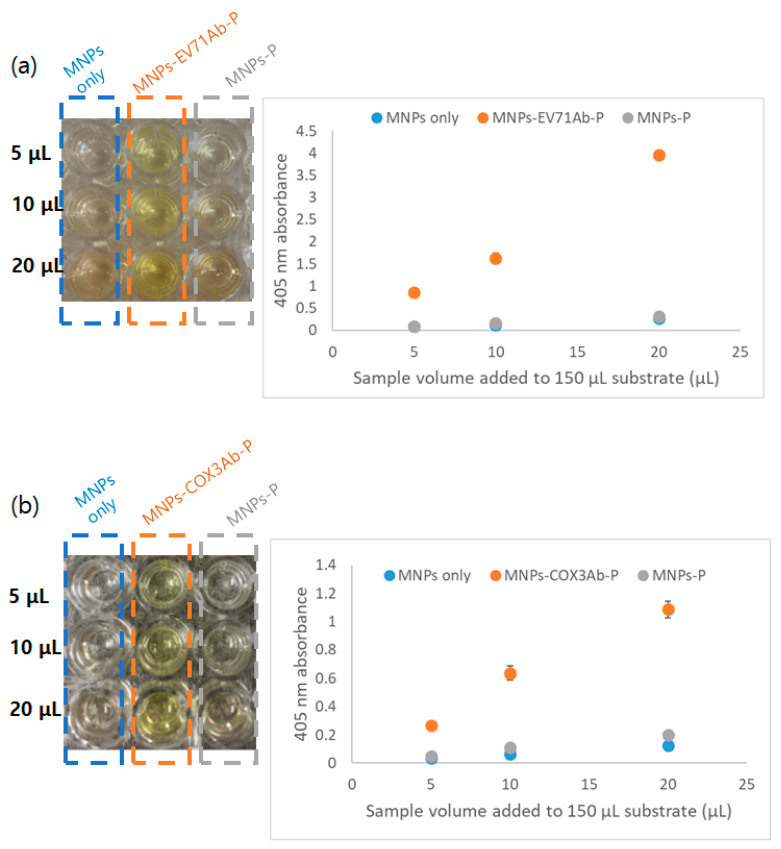
Confirmation of antibody binding of (**a**) EV71Ab and (**b**) COX3Ab antibody-conjugated nanoparticles utilizing the phosphatase method. The left images in (**a**,**b**) depict yellowish color expression photographs in a 96-well plate, while the right plots represent 405 nm UV/Vis absorbance. MNPs, MNPs-EV71Ab(COX3Ab)-P, and MNPs-P refer to bare MNPs, antibody-conjugated MNPs after incubating with a secondary alkaline phosphatase antibody, and bare MNPs after incubating with a secondary alkaline phosphatase antibody, respectively.

**Figure 3 diagnostics-13-03033-f003:**
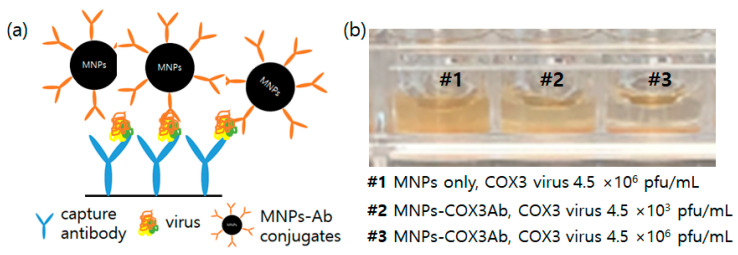
Color development due to aggregate formation of MNPs (**a**) Schematic diagram (**b**) Picture of bare MNPs (#1) and COX3Ab-conjugated MNPs (#2, #3) in the presence of COX3 virus in a 96-well plate.

**Figure 4 diagnostics-13-03033-f004:**
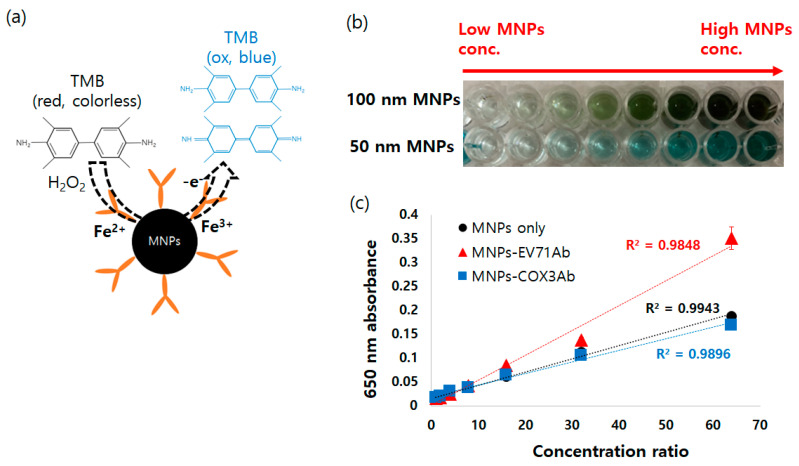
Color development due to peroxidase-like activity of MNPs (**a**) Mechanism of intrinsic peroxidase-like activity of MNPs (**b**) Color development of 100 nm and 50 nm MNPs (**c**) 650 nm absorbance of bare MNPs and antibody-conjugated MNPs under TMB substrate solution and H_2_O_2_ solution.

**Figure 5 diagnostics-13-03033-f005:**
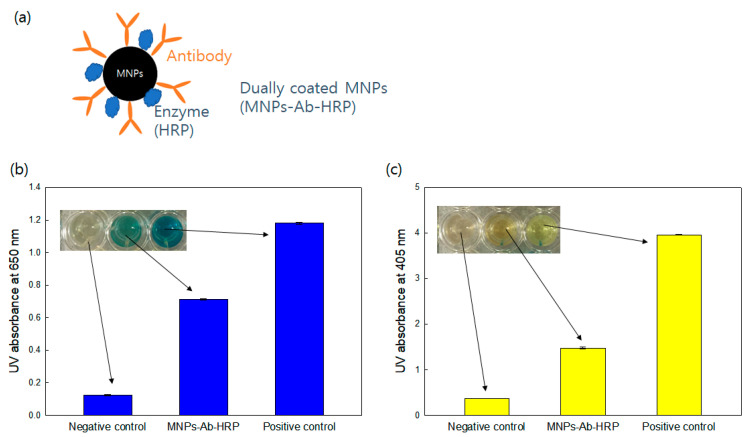
MNPs conjugated with antibody and HRP and their confirmation (**a**) Schematic illustration of the MNPs coated with antibody and HRP (MNPs-Ab-HRT) (**b**) Confirmation of HRP conjugation on MNPs utilizing TMB substrate solution with H_2_O_2_ (Negative control: PBS only, Positive control: HRP) (**c**) Confirmation of antibody conjugation on MNPs with the phosphatase method (Negative control: PBS only, Positive control: alkaline phosphatase antibody).

**Figure 6 diagnostics-13-03033-f006:**
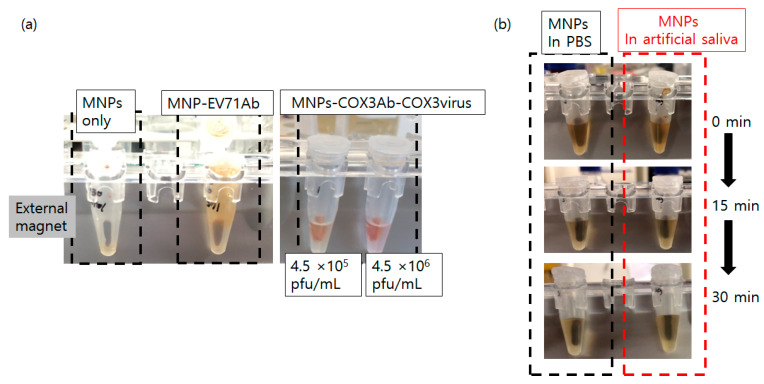
Concentration of MNPs using an external magnet (**a**) Concentration of MNPS (MNPs only), antibody-conjugated MNPs (MNP-EV71Ab), and COX3 virus-bound-antibody-conjugated MNPs (MNPs-COX3Ab-COX3virus) using an external magnet in PBS (**b**) Comparison of the concentration of MNPs using an external magnet field in PBS and artificial saliva solutions.

**Figure 7 diagnostics-13-03033-f007:**
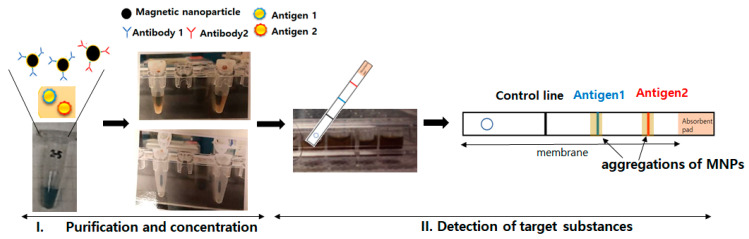
Two steps of rapid diagnostic tests (RDT) utilizing MNPs.

## Data Availability

Not applicable.

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
