# Peer review of "Exploiting the Potential of Magnetic Nanoparticles for Rapid Diagnosis Tests (RDTs): Nanoparticle-Antibody Conjugates and Color Development Strategies"

_diagnostics, 2023, doi:10.3390/diagnostics13193033_

Round 1

Reviewer 1 Report

This is a proof-of-concept project with an important justification in terms of the great potential for the use of nanoparticles.  Furthermore, the probes functionalised with antibodies, as proposed by the authors, can be used in different immunoassays, including ELISA and lateral flow immunochromatography. However, the work lacks rigour in the characterisation of the nanopaticles. The study could have been improved by tests to assess biological fluids, contaminated or not with the analytes of interest, to evaluet whether the immuno test is affected by the presence of the different components of plasma, serum, etc. There are no tests to validate specificity, thermal and functional stability. As  well as the functionality of the nanoparticles over time (days, weeks, months, etc). 

Author Response

I really appreciate for considerable comments and constructive suggestions for improvement of the manuscript. I delineate below all the answers and changes made in response to the comments.

Point 1: This is a proof-of-concept project with an important justification in terms of the great potential for the use of nanoparticles.  Furthermore, the probes functionalised with antibodies, as proposed by the authors, can be used in different immunoassays, including ELISA and lateral flow immunochromatography. However, the work lacks rigour in the characterisation of the nanopaticles. The study could have been improved by tests to assess biological fluids, contaminated or not with the analytes of interest, to evaluet whether the immuno test is affected by the presence of the different components of plasma, serum, etc. There are no tests to validate specificity, thermal and functional stability. As  well as the functionality of the nanoparticles over time (days, weeks, months, etc). 

Response 1:

1) Characteristics of nanoparticles: The purpose of nanoparticle characterization in this study is twofold. One aspect is to determine the size of nanoparticle aggregations that unavoidably occur due to the use of solvents containing buffers or other substances. The other aspect is to confirm the synthesis of antibodies by comparing the size of nanoparticles before and after antibody conjugation. In this study, nanoparticle-antibody conjugation was synthesized through repetitive iterations using the methods described in this article. Occasionally, when measuring the size using DLS, it was confirmed that the size variation was less than 10%, demonstrating the reproducibility of the method. Furthermore, to confirm antibody synthesis, a phosphatase method using a secondary antibody that binds only to the specific antibody was employed, making surface analysis of MNPs after MNPs-antibody conjugation unnecessary in my opinion.

2) Assessment of the effects of biological fluids: As suggested by the reviewer, I consider it highly important to investigate the impact of different biological fluids across the entirety of the immunoassay methodology. In relation to this, within this article, I have utilized an artificial saliva sample characterized by higher viscosity and diverse components compared to regular buffers(e.g. PBS), in the experiment involving the critical external magnetic field-assisted MNPs collection for the purification and concentration of target materials. The results demonstrate that the magnetic nanoparticles employed in this study can be collected by the external magnetic field within 30 minutes even when using artificial saliva samples. Presently, I am in the process of designing an actual immunoassay for virus detection as a follow-up to this study. Additionally, I plan to explore the influence of various biological fluids in the future.

3) Spcificity, thermal and functional stability of MNPs based immunoassays: The most significant advantage of the MNPs-based immunoassays proposed in this study is the potential to replace enzymes (e.g., HRP) typically used for color development. MNPs offer greater stability against thermal, chemical, and physical influences compared to enzymes, making them generally more suitable for use in immunoassays. As a result, it is anticipated that the MNPs-based immunoassay may exhibit improved specificity, thermal stability, and functional resilience compared to conventional immunoassays that rely on enzymes. However, it should be noted that the stability of nanoparticles can be affected by factors such as the formation of aggregations in the medium, leading to size enlargement over extended periods. Therefore, it appears that the utilization of MNPs in immunoassays should ideally take place within one week after synthesizing MNPs-antibody conjugations, as indicated in the revised manuscript.

Reviewer 2 Report

1- To confirm Figure 1 and DLS results, FESEM result should be provided for all the individual and composite materials.

2- Quality of figure 4 is very low.

3- On what basis is the concentration selected for antiviral tests? It seems that systematic optimization is necessary.

4- In the result and discussion section, it is not common to write the significant results. Please combine the sentences in one paragraph.

5- In section 3, both Results and Discussions were considered but in section 4, Discussion has been brought. Please remove discussions from section 3.

6- some related literatures were suggested to be cited for broader reader, such as: A) Ruiz-Pulido G, Medina DI, Barani M, Rahdar A, Sargazi G, Baino F, Pandey S. Nanomaterials for the diagnosis and treatment of head and neck cancers: A review. Materials. 2021 Jul 2;14(13):3706, B) A new approach to the development and assessment of doxorubicin-loaded nanoliposomes for the treatment of osteosarcoma in 2D and 3D cell culture systems." Heliyon 9, no. 5 (2023).

7- English writing need to edit by native person

Author Response

I really appreciate for considerable comments and constructive suggestions for improvement of the manuscript. I delineate below all the answers and changes made in response to the comments.

Point 1: To confirm Figure 1 and DLS results, FESEM result should be provided for all the individual and composite materials.

Response 1: In this study, the utilization of DLS for nanoparticle characterization serves a dual purpose. One aspect is to ascertain the size of nanoparticle aggregations that inevitably form due to the presence of solvents containing buffers or other substances. The other aspect is to validate the successful synthesis of antibodies by comparing the nanoparticle size before and after antibody conjugation. The process of nanoparticle-antibody conjugation was achieved through repetitive iterations emplying the methods outlined in this article.

               Periodically, during DLS measurements, it was observed that the size variation was within 10%. Additionally, it was evident that antibody-conjugated nanoparticles exhibited a notably larger size compared to bare nanoparticles. This observation lends support to the assertion that DLS measurements are both reasonable and reproducible. Drawing from my externsive experience in analyzing nanoparticle sizes using both DLS and TEM (Ha et al., 2015; Ha et al., 2016; Ha et al., 2022), I have concluded that the disparities in size measurements of nanoparticles between DLS and TEM analyses are not significantly substantial. This conclusion is also supported by prior studies (Khashayar et al., 2017; Chen et al., 2019). Particularly, I posit that capturing microscopic images in conjunction with DLS analysis would not significantly enhance the assessment of antibody conjugation.

               Regarding the reviewer’s comment, it’s worth noting that the size of magnetic nanoparticles derived from DLS measurements is slightly larger (by a few to tens of nanometers) compared to the size determined from microscopic images due to the formation of a hydration shell. However, I believe this difference may not be easily discernible by comparing DLS data with TEM images. Furthermore, this disparity is not likely to have a substantial impact on confirming the occurrence of antibody-MNP conjugation.

Ha et al., 2015, Distrubution of fullerene nanoparticles between water and solid supported lipid membranes: Thermodynamics and effects of membrane composition on distribution, Environmental Science and Technology 49(24), 14546-14553.

Ha et al., 2016, Bioavailability of fullerene under environmentally relevant conditions: Effects of humic acid and fetal bovine serum on accumulation in lipid bilayers and cellular uptake, Environmental Science and Technology 50(13), 6717-6727.

Ha et al., 2022, Elucidating the mechanism of cellular uptake of fullerene nanoparticles, Environmental Engineering Research 27(2), 200658

Khashayar et al., 2017, Fabrication and Verification of Conjugated AuNP-Antibody Nanoprobe for Sensitivity Improvement in Electochemical Biosensors, Scientific Reports, 7, 16070.

Chen et al., 2019, Self-Assembled Au@Fe Core/Satellite Magnetic Nanoparticles for Versatile Biomolecule Functionalization, ACS Applied Materials and Interfaces, 11, 23858-23869.

Point 2: Quality of figure 4 is very low.

Response 2: The quality of Figure 4 has been improved in the revised manuscript.

Point 3: On what basis is the concentration selected for antiviral tests? It seems that systematic optimization is necessary.

Response 3: As shown in Figure 3, the maximum concentration of the virus used in the experiment is 4.5 x 106 pfu/mL, which is the highest concentration feasible based on the virus stock available. Additionally, a 1000-fold dilution of 4.5 x 103 pfu/mL was used to assess the effect of virus concentration. This concentration range is consistent with the concentrations used in previous studies that investigated immunoassays using the same virus. For instance, a study by Wang et al.(2015) used EV71 concentrations ranging from 3.0 x 103 to 4.0 x 104 pfu/mL, and COX3 virus concentrations ranging from 2.0 x 103 to 3.0 x 104 pfu/mL. Furthermore, the concentrations of other chemicals, including the nanoparticles used in this study, were carefully selected based on preliminary tests and literature research.

Wang et al., 2015, Simultaneous point-of-care detection of enterovirus 71 and coxsackievirus B3. Anal. Chem. 87, 11105-11112.

Point 4: In the result and discussion section, it is not common to write the significant results. Please combine the sentences in one paragraph.

Response 4: In this article, Section 3 presents the main research results and their interpretations, while Section 4 discusses the key findings of this study and proposes the design of an immunoassay based on these findings. Therefore, the title of Section 3 has been changed to "Results," and the title of Section 4 has been changed to "Discussion."

Point 5: In section 3, both Results and Discussions were considered but in section 4, Discussion has been brought. Please remove discussions from section 3.

Response 5: Following the reviewer’s comment, I have removed the discussions from section 3.

Point 6: some related literatures were suggested to be cited for broader reader, such as: A) Ruiz-Pulido G, Medina DI, Barani M, Rahdar A, Sargazi G, Baino F, Pandey S. Nanomaterials for the diagnosis and treatment of head and neck cancers: A review. Materials. 2021 Jul 2;14(13):3706, B) A new approach to the development and assessment of doxorubicin-loaded nanoliposomes for the treatment of osteosarcoma in 2D and 3D cell culture systems." Heliyon 9, no. 5 (2023).

Response 6: Thank you for providing valuable references. I have cited two of the references in the revised manuscript.

Point 7: English writing need to edit by native person

Response 7: I have carefully edited the English writing throughout the revised manuscript.

Reviewer 3 Report

This paper has synthesized and validated antibody-conjugated MNPs were successfully through the measurement of MNPs' size and the use of phosphatase methods. It has also developed color in MNPs-based immunoassay via formation of MNP aggregations, MNPs' peroxidase-like activity, and synthesis of dually conjugated MNPs with both enzyme and antibody. The effective concentration and purification of target substances were also demonstrated.

Overall, this manuscript is of practical value. However, it is difficult to understand the mechanism of color development. I can recommend it for publication if the authors can explain more clearly why different colors can develop for different MNP concentrations. The authors are suggested to refer to the following papers:

1.  Jin et al. Nanocomposite coatings with plasmonic structural colors for subambient daytime radiative cooling. Solar Energy, 240: 211-224, 2022

2. Xiao et al.Bioinspired bright noniridescent photonic melanin supraballs. Science Advances,  3 (9), e1701151, 2017.

Author Response

I really appreciate for considerable comments and constructive suggestions for improvement of the manuscript. I delineate below all the answers and changes made in response to the comments.

Point 1: This paper has synthesized and validated antibody-conjugated MNPs were successfully through the measurement of MNPs' size and the use of phosphatase methods. It has also developed color in MNPs-based immunoassay via formation of MNP aggregations, MNPs' peroxidase-like activity, and synthesis of dually conjugated MNPs with both enzyme and antibody. The effective concentration and purification of target substances were also demonstrated.

Overall, this manuscript is of practical value. However, it is difficult to understand the mechanism of color development. I can recommend it for publication if the authors can explain more clearly why different colors can develop for different MNP concentrations. The authors are suggested to refer to the following papers:

  1. Jin et al. Nanocomposite coatings with plasmonic structural colors for subambient daytime radiative cooling. Solar Energy, 240: 211-224, 2022
  2. Xiao et al.Bioinspired bright noniridescent photonic melanin supraballs. Science Advances,  3 (9), e1701151, 2017.

Response 1: Thank you for recommending two valuable papers with insights on the color expression of nanoparticles. These two papers have been cited in the revised manuscript for a general explanation of color generation by nanoparticles. In the sections discussing MNPs aggregations (section 3.2.1) and peroxidase-like activity (section 3.2.2), the phenomenon wherein color expression becomes more pronounced as the concentration of nanoparticles increases has been elucidated for better clarity in the revised manuscript. They read:

  • It is generally acknowledged that the coloration of nanoparticles stems from either the absorption of light by the particles or the scattering of light through nanostructures [32,33].
  • Furthermore, as the virus concentration increases, more nanoparticles settled at the bottom, creating a denser yellow color.
  • Furthermore, under the optimized conditions, the 650 nm absorbance was directly proportional to the concentration of MNPs in the concentration range of 0 to 3.5 mg mL-1 (Figure S1(b) and Figure S1(c)). This is because as the number of nanoparticles increases, a greater amount of the oxidation form of TMB is generated.

Reviewer 4 Report

The author considers applications of magnetic nanoparticles in immunoassays (namely, in ELISAs and lateral flow tests). The author reasonably gives refs. 18-26 as examples of earlier works in the same field. The declared specific features of the presented study are proposed techniques for confirmation of the nanoparticle-antibody conjugation, three methods of color development and strategies for their applications in rapid diagnosis tests. The manuscript presents original experimental data, but their consideration and integration into new, more efficient analytical techniques is very poor. The study needs significant experimental extension, and the manuscript needs conceptual rewriting to make the work acceptable for publication in the Diagnostics journal or even to several publications. As well as such improvements should be associated with major changes and a large experimental refinement, the actually submitted work should be rejected.

See below the reasons for this decision and main comments about the proposed improvements of the manuscript:

1. The Abstract should contain short description of specific developments that were realized by the author including the reached improvements. Actually the Abstract only declares that the work in the chosen direction will be associated with some unclear progress.

2. The author indicates color development strategies in the title as the main direction of the presented development. However, the research is described in the manuscript almost without this term. Line 58 repeats the title and line 277 indicates Fig. 5a as a presentation of one strategy. The proposed strategies should be presented as some list determining the following structure of the paper or the term «strategies» should be excluded.

3. The proposed techniques need more focused comparison along all the manuscript with their known closest analogs. Without such discussion the opinions of the author about preference of the proposed techniques are not grounded.

4. The presented study considers application of already known carbodiimide technique (refs. 13,27) for conjugation of antibodies with commercial MNPs. So the risks on non-successful conjugation are limited and the stated task to confirm the conjugation is not actual. Really an important question is more detailed characterization of the obtained conjugates, i.e. confirmation of high degree of the conjugation, for example by finding composition of the conjugates. Unfortunately, the techniques selected by the author in its actual form are not acceptable for this purpose.

5. The author selects only two approaches for characterization of the prepared conjugates excluding their microscopic studies, testing changes of surface properties, etc. The reasons of this choice are not grounded.

6. The consideration of DLS data causes serious questions. The manufacturer states that average diameters of its MNPs are 50 nm and 100 nm (see lines 67-68). However, Section 3.1.1 considers the size changes starting from 190.40 nm. This difference is interpreted as aggregation (lines 171-175), but the manuscript does contain direct demonstration of this process (for example, by microscopy). High reproducibility of DLS data (RSD < 1%, see line 169) is not expected for complexes of a small number of homogeneous particles. Besides, a well-known fact is overestimation of particle sizes in DLS measurements due to taking into account their fixed nearest environment, such as the hydration shell. So the consideration of DLS technique needs essential revision.

7. The author states the use of MNPs aggregation as the proposed approach for color development (see Section 3.2.1), but the manuscript does not contain spectral comparison of initial and aggregated MNPs. Are the proposed aggregates stable sufficienly for reproducible assays? Will their structure change irreproducibly upon contact with biosamples? What is the proposed complete protocol of the use of the aggregated MNPs in the assay? Does the assay include such common property and advantage of MNPs as their ability to concentrate bound molecules in a small volume using magnetic field? There is no calibration curve for the technique presented in Fig. 3. The advantages of MNPs are not grounded by comparison with other colored labels.

8. The text in the Section 3.2.2 describes peroxidase-like activity of MNPs addressing to the work published in 2007 (lines 237-238). However, a lot of nanozymes, i.e. nanoparticles with enzyme-like activities are actually known, as well as examples of their applications in immunoassays. The advantages of the used MNPs as peroxidase-like catalyst are not demonstrated in the manuscript. Besides, dextran coverage limits availability of the surface of the nanoparticles for contacts with the substrate. So the choice of efficient MNPs also is a task to be solved. Fig. 4 and its discussion are limited by homogeneous testing of MNPs as catalysts and do not provide complete assay protocol with the possibility to compare its detection limit with properties of already known techniques.

9. The data in the Section 3.2.3 cause the same questions. The author does not provide complete assay scheme. The reasons to use MNPs in it are not specified and confirmed. The microplate immunoassays may be implemented by direct antibody coupling to HRP, to available polymeric preparations of HRP or to other carriers such as gold nanoparticles. The presented data does not contain any concentration dependence for the detection of some analyte.

10. Concentration of target compounds by magnoimmunosorbents is described in a row of earlier works. Specific features of the author’s proposition are not stated in the manuscript, and their advantages are not confirmed experimentally. The presented results (Fig. 6) are limited by conceptual demonstration of the already known effect of concentration using an external magnetic field. The paper does not contain complete assay protocol and reached analytical parameters of this assay.

11. Fig. 7 is limited by the demonstration of already presented effect of MNPs-based concentration and schemes of lateral flow test. There are no any experimental data demonstrating the author's results with such tests. Note that the integration of MNPs as concentrating tools and colored labels with lateral flow tests is already known from literature.

12. Sections 4 and 5 repeat already stated declarations about analytical applications of MNPs, but do not contain new data or meaningful comparisons with other works to evaluate the proposed approaches.

Round 2

Reviewer 3 Report

The authors have addressed my comments. I can recommend it for publication.

Author Response

I sincerely appreciate your recommendation for this paper to be published. 

Reviewer 4 Report

The author has revised the manuscript and prepared detailed answer to the reviewer that clarify the author's position. However, new submitted version of the manuscript is still characterized by significant gaps in the presented experimental material, the absence of (i) characterized complete analytical developments and (ii) confirmation of their advantages. In this regard, I still believe that the manuscript should be rejected.

Author Response

This paper aims to (i) present the appropriate method for the antibody conjugation of 100 nm dextran coated magnetic nanoparticles(MNPs) and its validation method, and (ii) experimentally demonstrate the feasibility of various color representations using MNPs-antibody conjugates, and proposed a rapid detection kit based on these concepts. In line with the research objectives, analytical development can be summarized into two main aspects. Firstly, a clear protocol has been provided for the antibody conjugation of 100 nm dextran-coated MNPs and the means to confirm this process. Secondly, I have presented various methods for implementing color representations with MNPs in the diagnostic kit, along with precise protocols and demonstrated feasibility for each implementation. This paves the way for potential applications in diagnosing various target materials in the future.

The confirmation of the advantages of the analytical developments in this research can be summarized as follows. Firstly, it clearly demonstrates the potential for quantitative analysis by showing that color expression is proportional to nanoparticle concentration or virus concentration in all color implementation methods. Secondly, it illustrates the ease of implementing effective purification and concentration of MNPs using an external magnet, regardless of factors such as antibody conjugation, dispersion medium, or virus binding. This establishes the feasibility of detecting target materials in various biological samples (e.g., buffer and saliva) or matrices in the future.

 As shown in the response in Round 1, I am currently conducting further research to implement a complete assay using the color expression methods mentioned described in this paper. I am also working on improving aspects such as nanoparticle size to yield more significant research outcomes. Additionally, I greatly appreciate the meaningful comments provided during Round 1, which will be of significant assistance not only for this paper but also for our ongoing further study.